# Unveiling the spectrum of Arabic offensive language: Taxonomy and insights

Chaya Liebeskind[1]*, Yossef Haim Shrem[1], Marina Litvak[2], Natalia Vanetik[2]

**1** Department of Computer Science, Jerusalem College of Technology, Jerusalem, Israel, **2** Department of Software Engineering, Shamoon College of Engineering, Beer-Sheva, Israel

* liebchaya@gmail.com

**Data availability statement:** All data is available from the GitHub repository: https://github.com/NataliaVanetik/ArabicOffensiveLanguage_TaxonomyAndData.

## Abstract

This paper presents a novel taxonomy designed to classify offensive language in Arabic, filling a notable void in existing literature primarily concentrated on Indo-European languages. Our taxonomy delineates offensive language into seven distinct levels, comprising six explicit levels and one implicit level. Drawing inspiration from the simplified offensive language (SOL) taxonomy outlined in prior work, we adapted it to accommodate the intricacies and linguistic richness of Arabic. In our study, we analyzed existing datasets containing offensive language in Arabic, examining the range of annotations employed within these datasets. This exploration allowed us to gain insights into the diversity of offensive language instances and the methodologies used for their annotation, thereby informing the development of our streamlined taxonomy for categorizing such expressions. Initial examination of datasets uncovers compelling trends and distributions, emphasizing the intricate and distinct nature of offensive expressions in Arabic. We have also analyzed the performance of pre-trained and fine-tuned Arabic transformer offensive language detection models on these datasets. Our results underscore the importance of acknowledging linguistic and cultural diversity in the study and mitigation of online abusive language. We posit that our refined taxonomy and accompanying dataset will be pivotal in advancing research across Semitic languages, including sociocultural studies, natural language processing, and linguistic analyses.

## 1. Introduction

The expanding field of internet communication, especially in Arabic, calls for effective measures to uphold decorum and shield users from offensive material. In order to accomplish these goals, automatic offensive language identification is essential. While significant advancements have been made in Arabic offensive language detection using deep learning models like pre-trained and fine-tuned transformers, a standardized approach to data annotation and classification remains elusive. This gap hinders the development of truly comprehensive detection systems. Our work addresses this challenge by introducing a tailored taxonomy for the Arabic offensive language, which is the cornerstone for effective dataset analysis.

This taxonomy employs a hierarchical structure with clearly defined categories, guiding the process of consistent data labeling and facilitating the development of robust detection

**Funding:** Chaya Liebeskind Israel innovation authority https://innovationisrael.org.il/en/. The funders had no role in study design, data collection and analysis, decision to publish, or preparation of the manuscript.

**Competing interests:** The authors have declared that no competing interests exist.

models. The taxonomy tackles the complexities of Arabic offensive language, acknowledging the influence of cultural nuances and dialectal variations.

To understand how effectively existing datasets capture these complexities, we conduct a thorough analysis of 17 prominent datasets. By aligning these datasets with our proposed taxonomy, we assess how well they cover the various categories of offensive language in Arabic. This analysis identifies potential gaps and biases within existing datasets, informing future data collection efforts. We further explore the compatibility of these datasets with each other, particularly in terms of dialectal variations. We evaluate the potential for merging certain datasets or creating a new, comprehensive dataset that reflects the full spectrum of Arabic offensive language.

Furthermore, we investigate the efficacy of existing deep learning models for Arabic offensive language detection, with a specific focus on pre-trained and fine-tuned transformers. We examine how well these models perform when applied to datasets from different dialects, building upon our analysis of dataset compatibility. As a result, we can evaluate the transferability of learned features necessary for efficient transfer learning and the potential need for dialect-specific models.

By combining a novel taxonomy with a comprehensive dataset analysis and an exploration of existing deep learning models, including pre-trained and fine-tuned transformers, this paper aims to: (1) improve the accuracy and robustness of Arabic offensive language detection systems; (2) identify areas for improvement in existing datasets and explore the possibility of merging them; (3) encourage the development of dialect-specific models where necessary.

There are several real-world uses for offensive language detection in the Arabic language, such as content moderation, online community management, and social media surveillance. We strive to improve the accuracy of automated offensive detection systems by creating a uniform taxonomy and joining multiple existing datasets. Furthermore, considering the wide linguistic variety of the Arabic language in different countries, our work can facilitate the creation of dialect-specific detection models, which are currently underdeveloped.

This paper introduces a novel offensive language taxonomy tailored for Arabic, which enhances the granularity and relevance of dataset annotations (Sect 3). We provide a detailed description of each level of the taxonomy, highlighting its significance in improving dataset analysis and facilitating cross-dialectal studies (Sect 4). Our analysis covers 17 datasets, including re-annotated and filtered versions, and introduces a new combined dataset for Modern Standard Arabic (MSA) and Levantine dialects (Sect 5.3.3).

To improve model accuracy, we evaluate pre-trained and fine-tuned Arabic transformer models, applying techniques such as transfer learning across dialects and fine-tuning on re-annotated datasets to capture nuanced patterns in offensive language (Sect 5.2). Our experiments demonstrate the efficacy of combining dialect-specific datasets with the proposed taxonomy in enhancing classification performance. Finally, we discuss the broader implications of our findings for advancing research in Arabic offensive language detection and outline potential directions for future work (Sect 6).

## 2. Related work

### 2.1. Offensive language detection in Arabic

With the global proliferation of social media as a communication platform, the challenge of multilingual content analysis has gained prominence, particularly in offensive language detection across diverse languages, including Arabic [1]. Researchers have responded to this challenge by developing innovative methodologies and annotated corpora, which have evolved over time.

Initial research on Arabic offensive language detection focused on leveraging traditional machine learning and deep learning methodologies. Husain et al. [2] introduced Salam-NET, a Bi-directional Gated Recurrent Unit (Bi-GRU)-based system, for the SemEval 2020 Multilingual Offensive Language Identification shared task. This foundational work demonstrated the efficacy of recurrent neural networks for Arabic text classification. Concurrently, Husain [3] conducted a comprehensive review of 35 studies, identifying significant gaps in Arabic offensive language detection, particularly in areas such as cyberbullying and adult content. Addressing these gaps, Husain [4] showed the advantages of ensemble learning using datasets from the OSACT workshop at LREC 2020.

Building on these foundational efforts, researchers began exploring the capabilities of deep neural networks (DNNs) and large language models (LLMs). Haddad et al. [5] introduced a hybrid model combining Convolutional Neural Networks (CNNs) with Bidirectional GRUs and attention mechanisms, achieving superior results on Arabic datasets. Around the same time, Althobaiti [6] compared BERT against traditional classifiers like SVM and logistic regression, showcasing BERT's exceptional performance in detecting offensive language and fine-grained hate speech categories such as race and religion. These studies marked a significant shift toward transformer-based approaches, emphasizing their potential for handling complex linguistic nuances.

Recent studies have focused on optimization techniques to improve classification accuracy and robustness. Shannaq et al. [7] proposed a two-stage optimization pipeline incorporating pre-trained word embeddings, XGBoost, SVM, and genetic algorithms. Their approach, tested on the Arabic Cyberbullying Corpus (ArCybC), excelled across diverse domains like gaming and news. Khairy et al. [8] demonstrated the effectiveness of ensemble models over single learners for offensive language detection, achieving superior results across multiple datasets.

Dialectal Arabic presents unique challenges due to its linguistic diversity and mixed-script usage. Boucherit and Abainia [9] focused on detecting offensive content in Algerian dialectal Arabic, tackling the complexities of Arabizi (mixed Roman and Arabic script) and multilingual influences. Husain and Uzuner [10] explored transfer learning across datasets, revealing its potential limitations for highly dialectal content. In response, Aljuhani et al. [11] utilized BiLSTM models with domain-specific word embeddings, achieving 93% accuracy on Arabic Twitter datasets.

To address the scarcity of resources, cross-lingual approaches have emerged as a promising direction. Litvak et al. [12] conducted experiments with Arabic and Hebrew, demonstrating that transfer learning across Semitic languages can improve classification performance. This work underscores the importance of leveraging linguistic similarities to overcome data limitations.

From the papers surveyed, it can be observed that the evolution of Arabic offensive language detection research reflects a steady progression from traditional classifiers to advanced neural and transformer-based architectures, with a growing focus on optimizing for dialects, scripts, and resource scarcity.

The table summarizing these works is available in the Appendix.

## 2.2. Arabic offensive language datasets

The development of datasets for offensive language detection in Arabic has been pivotal in addressing abusive online behavior, especially in the linguistically and culturally diverse Arab-speaking regions. These efforts provide the foundation for training and evaluating models to combat hate speech and abusive language on social media platforms. Below, we categorize and highlight significant works based on their focus and contributions.

Albadi et al. [13] targeted religious hate speech in Arabic tweets, creating a dataset of 6,136 manually annotated tweets aimed at identifying biases against religious groups. Similarly, Aref et al. [14] presented a dataset tailored for detecting religious hate speech in Arabic, particularly addressing the Sunni-Shia divide. Their dataset contains 3,235 annotated tweets. Expanding on specific themes, Mulki et al. [15] introduced the Let-Mi dataset, designed for identifying misogynistic language in Levantine Arabic tweets during the October 17th protests in Lebanon. Annotators categorized the data into eight types of misogynistic behavior, including sexual harassment and stereotyping.

Several works have focused on creating large-scale general-purpose datasets. Mubarak et al. [16] constructed a dataset of 660,000 Arabic tweets, of which 10,000 are annotated, focusing on identifying offensive content across various dialects. This dataset was used in the OffenseEval workshop [17]. Similarly, Haddad et al. [18] developed the T-HSAB dataset, containing 6,039 messages in the Tunisian Arabic dialect tagged for abusive and hate speech. More recently, the Hate Speech Dataset for the Saudi Dialect [19] introduced 23,000 tweets annotated for hate speech, with data collected over three months in 2023, emphasizing regional dialects and code-mixing.

Some datasets address more complex annotations with multidimensional frameworks. Ousidhoum et al. [20] presented a multilingual dataset (including 3,353 Arabic samples) annotated for hostility, directness, and target attributes such as religion or sexual orientation. Ahmad et al. [21] extended this concept with a multi-class dataset of 403,688 Arabic tweets categorized into four sentiment-based hate speech classes, ranging from severe hate speech to non-offensive language.

Researchers have also explored platform-specific datasets. Alakrot et al. [22] collected over 167,000 YouTube comments, focusing on annotating a subset for offensive language. Chowdhury et al. [23] presented a dataset of 4,000 dialectal Arabic news comments, annotated for offensive and vulgar content. Mubarak et al. [24] added another dimension with two datasets: one with 1,100 tweets and another with 31,692 comments from Al Jazeera's website, focusing on moderated abusive content.

Several works addressed offensive language in different Arabic dialects. Mulki et al. in [25] presented the L-HSAB, a dataset that focuses on hate speech and abusive language in the Levantine Arabic dialect on Twitter. The dataset contains 5,846 tweets. To address the scarcity of labeled data for underrepresented dialects, Litvak et al. [12] combined existing datasets [16,26] with new annotations, creating a corpus of 9,193 Twitter comments. Abdelhakim et al. [27] introduced the Ar-PuFi dataset, targeting offensive language against public figures in the Arab world. A dataset, particularly created for identifying abusive language in Moroccan Darija, the colloquial Arabic dialect used in Morocco, was introduced in [28].

This wide array of datasets highlights the evolving landscape of Arabic offensive language detection. While significant progress has been made, gaps remain in addressing underrepresented dialects, domain-specific contexts, and nuanced hate speech categories.

## 2.3. Taxonomies of offensive language

Taxonomies of offensive language for various languages, such as English, Czech, Lithuanian, Polish, and Hebrew, were introduced in previous works. Our work adopts the same approach to data annotation and evaluation while considering cultural and morphological differences between Arabic and these languages.

Lewandowska-Tomaszczyk et al. [29] presented a Simplified Offensive Language (SOL) Taxonomy, applied and tested in the Second Annotation Campaign between March and May 2023 on English, Czech, Lithuanian, and Polish. This taxonomy was verified and located in

Linguistic Linked Open Data (LLOD). Referring to previous taxonomic models proposed by the COST Action Nexus Linguarum WG 4.1.1 team (see https://nexuslinguarum.eu), the authors revised the definitions and variety of categories and tested the present typology on publicly available datasets for each language. They proposed this taxonomy as a core ontology for encoding offensive languages and justified its use on new data within a more universal LLOD schema.

Following previous study, Lewandowska-Tomaszczyk et al. presented an integrated model of explicit and implicit offensive language taxonomy in [30]. The study relied mainly on categories proposed by Zampieri et al. in [31]. In the analytic procedure, they first distinguished offensive discourse from non-offensive discourse, with offense targets and subsequent categorization levels identified. The authors proposed the concept of offensive language as a superordinate category in the system with 17 hierarchically arranged subcategories. The categories are structured taxonomically into four levels and verified using neural-based (lexical) embeddings. Together with a taxonomy of implicit offensive language and its subcategorization levels, which have received little scholarly attention until now, the categorization is exemplified in samples of offensive discourse from selected English social media materials, including 25 publicly available hate speech datasets. The study discussed the offensive category levels (types of offense, targets, etc.) and aspects (offensive language property clusters) and proposed a computationally verified integrated explicit and implicit offensive language taxonomy.

Extending the above studies, Liebeskind et al. introduced a streamlined taxonomy for categorizing offensive language in Hebrew in [32], addressing a gap in the literature that previously focused largely on Indo-European languages. This taxonomy divides offensive language into seven levels (six explicit and one implicit). It is based on the simplified offensive language (SOL) taxonomy introduced by Lewandowska-Tomaszczyk in [33], aiming to reflect the unique linguistic and cultural nuances of Hebrew. The authors manually analyzed the nuances of offensive language in Hebrew. We also constructed a dataset from Twitter and manually curated it to validate the taxonomy and highlighted the significance of considering linguistic and cultural variations when researching and correcting abusive language online.

## 3. Arabic simplified taxonomy of offensive language

The taxonomy of offensive language is crucial as it offers a systematic classification for various forms of offensive content. This, in turn, assists automated systems in effectively moderating and responding to such content. This taxonomy provides a fundamental framework that not only arranges the intricate landscape of online interactions but also acts as a tool to enhance the safety and usefulness of digital platforms by increasing understanding of online behavior. To establish a classification system for offensive language in Arabic, we modify and expand upon the taxonomy put forth by [30], which itself builds upon the taxonomy proposed by [31,34]. Hebrew [32], Czech, Lithuanian, and Polish [29] underwent the same procedure.

This taxonomy consists of seven fundamental levels, referred to as levels 1 to 7, for analysis. The initial six levels pertain to explicit categories, while the seventh level pertains to implicit categories. Explicit offensive language is characterized by its lack of ambiguity and directness, making it evident to the majority of individuals that the content is intended to provoke, insult, or cause harm. This includes explicit slurs, overt threats, openly racist or sexist statements, and similar expressions. Implicit offensive language refers to offensive content that is conveyed in a subtle or concealed manner, often necessitating an understanding of context or cultural knowledge to discern the offensive intent. Examples of implicit offensive

language include *inner-city schools* and *the welfare queen*. Here, we provide a comprehensive description of the levels.

1. **Level 1** distinguishes between *offensive* and *non-offensive* texts. Offensive texts are those that contain offensive language or a subject that, in certain contexts, may be deemed offensive. Non-offensive texts may be positive, neutral, or pertain to any subject matter.

2. **Level 2** designates the recipient of a derogatory discourse. The target can refer to an *individual*, a *group*, an *individual in relation to the group* (where stereotypes about the group are employed to offend someone), or other (when none of the aforementioned scenarios are applicable).

3. **Level 3** identifies whether the target of an objectionable speech is evidently identified and explicitly addressed in the text (in this instance, the target is *present*) or *absent* (the target is not participating in the conversation).

4. **Level 4** specifies whether the inappropriate language is *vulgar*, meaning it includes vulgar words, idioms, and curses, or non-vulgar, indicating the absence of vulgar terms or curses.

5. **Level 5** The offensive strength is determined by four categories, ranging from least severe to most severe, as shown below. .
   - An *insult* refers to a statement or behavior that is deliberately meant to offend or cause harm to someone by belittling them, displaying disdain, or ridiculing them. Insults should not rely on generalizations about certain groups.
   - *Hate speech* refers to the use of objectionable language that discriminates against, belittles, or encourages violent or prejudiced acts against an individual or a group based on characteristics such as race, religion, ethnic origin, sexual orientation, handicap, or gender (specifically by using generalizations about the group).
   - *Discrediting* someone involves damaging their positive reputation or creating doubt in others about their integrity, competence, or reliability by distributing false information, emphasizing actual or perceived weaknesses, or questioning their motives.
   - A *threat* is an indication that one intends to cause another person distress, harm, damage, or another negative outcome.

6. **Level 6** explores several facets of explicit offensive language, as outlined below. A solitary text has the potential to display one or more facets.
   - *Racism* – expressions that diminish, ridicule, categorize, or display bias against persons or groups according to their race or ethnicity.
   - *Xenofobia* – expressions that demonstrate a profound aversion, apprehension, or bias against individuals from other nations, cultures, or ethnicities.
   - *Homophobia* – exhibiting bias, contempt, apprehension, or animosity towards persons based on their sexual orientation, particularly against those who identify as gay, lesbian, or bisexual.
   - *Sexism* - utterances that diminish, ridicule, categorize, or display bias against someone based on their gender or sex.
   - *Profanity* – blasphemous phrases that demonstrate a lack of regard or scorn for religious ideas, persons, artifacts, or rituals.
   - *Ageism* – utterances that demean, ridicule, categorize, or display bias against persons depending on their age.
   - *Ableism* – expressions that demean, ridicule, categorize, or exhibit bias against persons due to their infirmities or perceived capabilities.
   - *Classism* – expressions that demean, ridicule, categorize, or display bias against persons based on their social, economic, or educational status.

- *Ideologism* – bias, prejudice, or discrimination based on political or ideological beliefs.
- *Other* – any feature that is not included in the aforementioned categories.

7. **Level 7** explores several facets of implicit offensive language, as delineated below. A solitary text has the potential to display one or more facets.
   - *Indirectness* – refers to the use of phrases or remarks that insinuate or indicate anything unfavorable or disparaging without explicitly stating it.
   - *Rhetorical questions* – inquiries that are asked without anticipating a response, but are intended to emphasize a certain point, generally in a sarcastic, disdainful, or patronizing tone.
   - *Simile* – a rhetorical device that draws a connection between two distinct objects using the words "like" or "as" to emphasize a certain trait or attribute. Similes may be used to diminish, ridicule, or degrade an individual or a collective by using unfavorable or uncomplimentary analogies.
   - *Metaphors* – figures of speech which include the use of words or phrases that are not relevant to an item or activity. These metaphors may be used to diminish, ridicule, degrade, or stereotype an individual, group, or idea by making unfavorable or unflattering analogies.
   - *Irony* – refers to the act of expressing anything that has a meaning that is opposed to its literal or customary meaning.
   - *Understatement* – a rhetorical device used by a speaker to deliberately diminish or trivialize the importance of something, therefore portraying it as less significant or serious than it is.
   - *Overstatement* (hyperbole) – a rhetorical device that involves exaggeration, often used to insult, disparage, or misrepresent someone or something in a contemptuous way.

Fig 1 Illustrates the taxonomy's levels 1-6, which pertain to explicit offensive language.

It is important to acknowledge that it might be more difficult to recognize offensive language that is not expressly spoken. Implicitly offensive language employs innuendos, euphemisms, or coded terminology that may not be readily seen as offensive unless the underlying context or meaning is comprehended. For example, a term might be considered neutral in one cultural or socioeconomic context but offensive in another. Understanding such nuances might be difficult. Therefore, this research specifically concentrates on the identification and categorization of explicit offensive words in the Arabic language. Thus, our only emphasis is on levels 1-6 of the aforementioned Arabic taxonomy. Due to the significant uncertainty and cultural dependence on implicit offensive language (level 7), we have deferred its examination to future work. This decision was made since it had a substantial impact on the annotation process.

The Arabic translation of this taxonomy is illustrated in Fig 2.

The translation of level 7 implicitly offensive terms is provided in Table 1.

Translating offensive language taxonomy is a delicate task that requires careful consideration for several reasons because different cultures have different norms and taboos. A term that is considered innocuous in one culture might be highly offensive in another. Failing to take this into account can lead to misunderstandings and unintended offenses. Direct translations can sometimes miss the nuance or weight of the original term. For example, Homophobia can be translated as رَهَابُ الْمِثْلِيَّة (fear of) or كُرَاب الْمِثْلِيَّة (hatred of).

The first translation is more faithful to the original meaning, while the second translation is more focused on offensive language and may imply a more aggressive approach. The presence of diglossia in Arabic, which refers to the use of two distinct dialects or languages in different

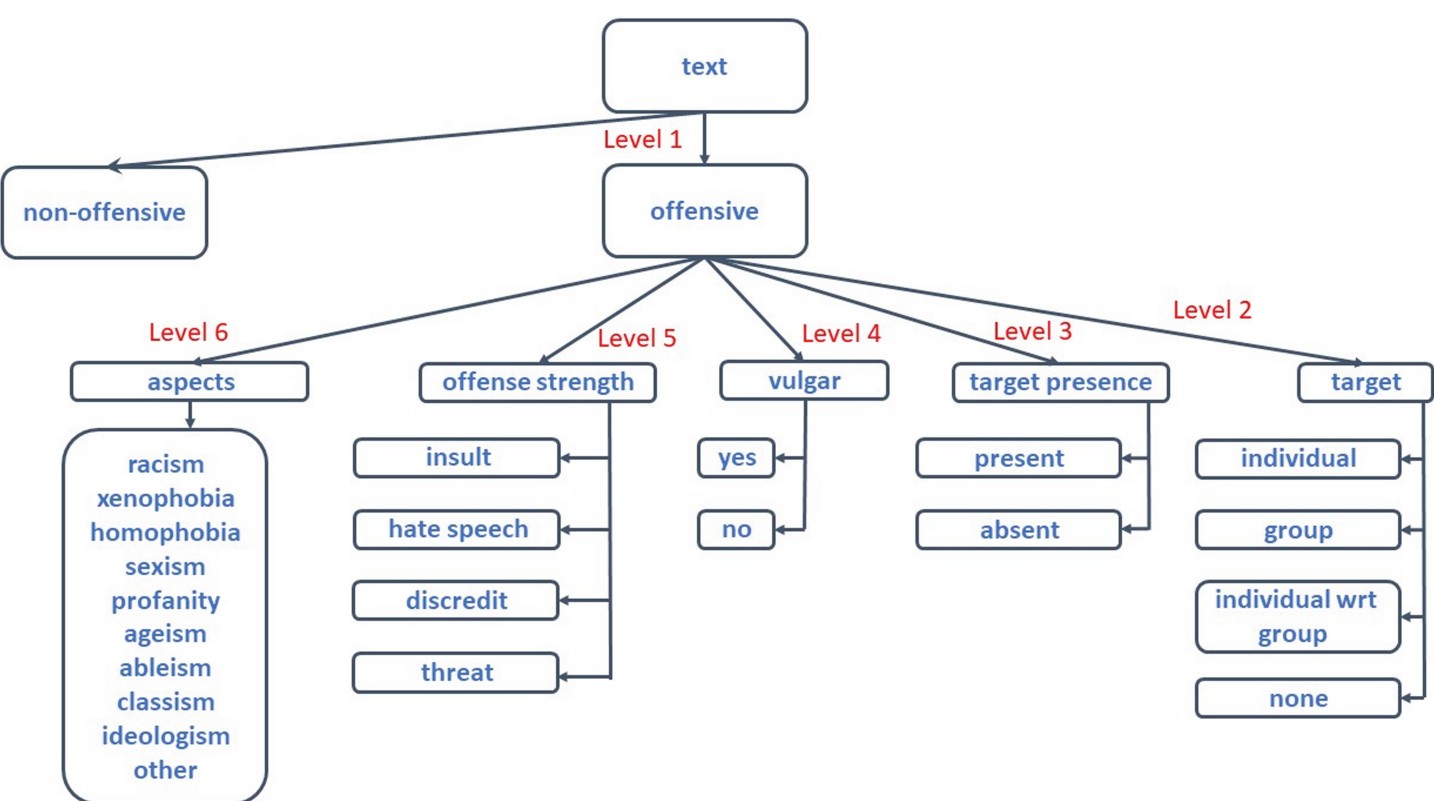

**Fig 1. Simplified offensive language taxonomy for English, levels 1-6.** The figure presents the taxonomy of explicit offensive language, organized into six levels. Level 1 differentiates between offensive and non-offensive texts. From right to left, Level 2 identifies the target of offense (individual, group, individual in relation to a group, or other). Level 3 specifies whether the target is present or absent in the text. Level 4 classifies the text as vulgar or non-vulgar. Level 5 defines the severity of offensiveness as insult, hate speech, discrediting, or threat. Level 6 enumerates specific aspects of offensive language.

social contexts by a single language community, adds an extra layer of difficulty to the translation of the taxonomy into English. We utilized professional dictionaries to achieve optimal translation of the taxonomy and sought the expertise of highly experienced linguists.

## 4. Arabic datasets for offensive language

In this section, we describe and analyze existing Arabic offensive language datasets and their annotations in the scope of the new Arabic offensive language taxonomy. The data is available at https://github.com/NataliaVanetik/ArabicOffensiveLanguage_TaxonomyAndData. We have evaluated 17 datasets published in the scientific literature between 2017 and 2024, focusing on various aspects of offensive language in Arabic, including hate speech, abusive language, and offensive comments across different social media – papers [12–28]. These datasets are manually annotated to ensure consistency and accuracy, covering different dialects, topics, and target attributes. The datasets vary in size, ranging from thousands to hundreds of thousands of text samples, providing a comprehensive resource for studying and detecting offensive language in Arabic.

The datasets cover a variety of dialects: 5 datasets contain texts written in Modern Standard Arabic (MSA), 3 datasets use Levantine Arabic, 4 datasets contain mixed Arabic dialects, and Gulf, Tunisian, Moroccan Darija, and Saudi dialects appear in one dataset each. Dataset sources also vary and cover a wide range of social media platforms. The most popular source

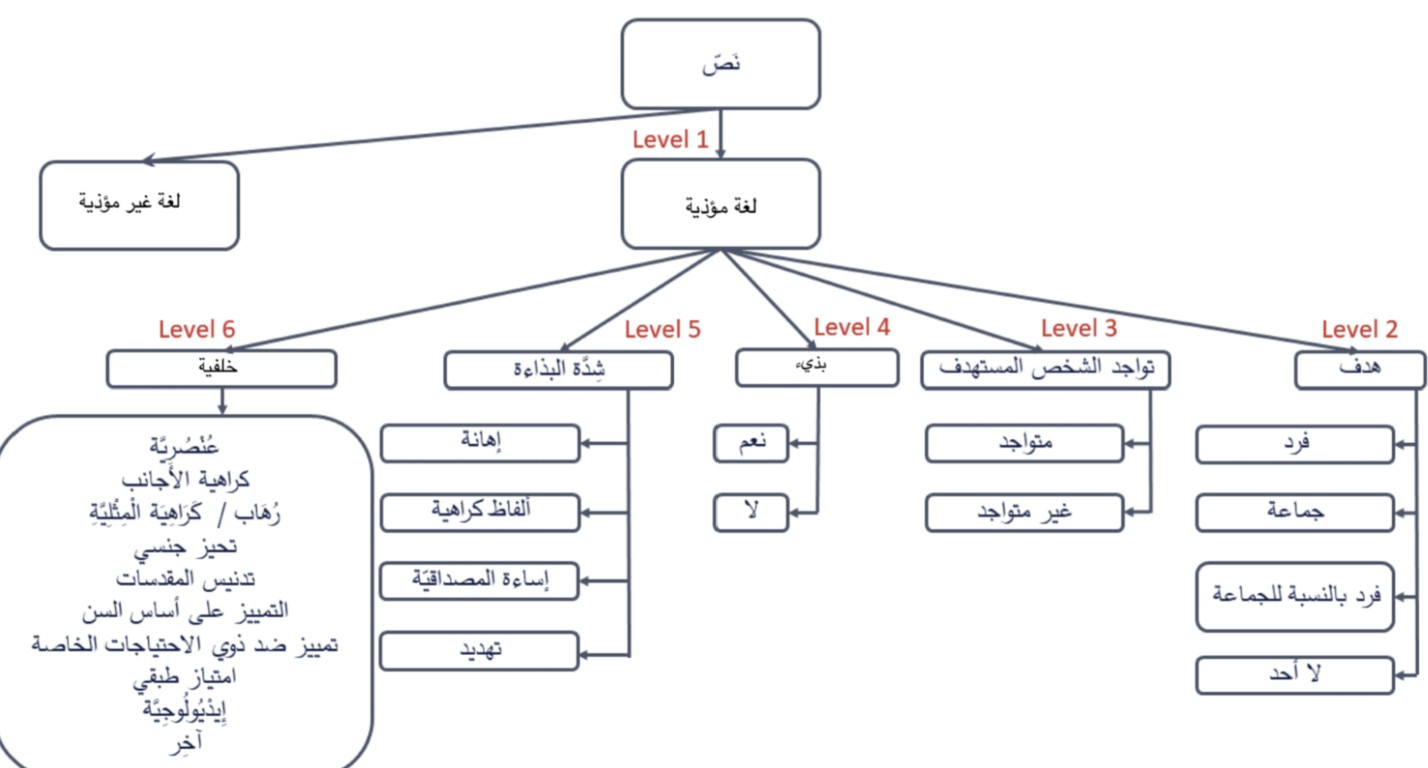

**Fig 2. Simplified offensive language taxonomy for Arabic.** This figure presents the Arabic version of the taxonomy outlined in Fig 1. It retains the six-level structure: distinguishing offensive from non-offensive texts (Level 1), identifying the target of offensive discourse (Level 2), specifying the target's presence or absence (Level 3), differentiating between vulgar and non-vulgar language (Level 4), categorizing the intensity of offensiveness (Level 5), and detailing specific aspects of offensive language, such as racism, xenophobia, and others (Level 6).

**Table 1. Arabic translation of implicit offensive language aspects (level 7).**

| Implicit aspect | Arabic translation |
| --- | --- |
| indirectness | عدم المباشر |
| rhetorical questions | أسئل بلاغ |
| simile | تشبِ |
| metaphor | مجَاز or ستِعار |
| irony | سخرِ ء or تَ رُم |
| understatement | تصرح م بوح |
| overstatement | مبَالَغ |

is Twitter (10 datasets), followed by YouTube (3 datasets), news websites (2 datasets), and mixed sources such as Twitter, Facebook, and YouTube (2 datasets). Themes and types of offensive language covered include religious hate speech (3 datasets), hate speech and abusive language in specific dialects (6 datasets), and general offensive and abusive language detection (7 datasets). All datasets analyzed in this study were collected from publicly available sources and used in accordance with their respective terms and conditions. The collection and analysis of the data comply with ethical research standards and are restricted to publicly accessible content intended for research purposes.

Below, we provide details on the papers introducing these datasets (full data analysis is performed in Sect 4.1).

## 4.1. Datasets description

This section describes annotations, size, and dialect of Arabic offensive language datasets listed in Sect 2.2. Table 2 presents information on 16 datasets we examine. We report a specific dialect or Modern Standard Arabic (MSA) for the datasets where it can be determined; in cases where a mixture of dialects is present in the data we report "undetermined".

## 4.2. Dataset filtering

Dataset 1 of [13] contains tweet IDs and does not contain the original texts. Because tweepy API [35] does not allow extraction of historical data from Twitter beyond one week [36], we do not proceed to examine this dataset.

Additionally, upon closer examination, we have observed that dataset 13 introduced [21] is not suitable for our purposes. Upon closer examination, we have found that texts labeled "positive" and "very positive" (offensive and deeply offensive) do not represent instances of offensive language. These texts do use some negative words, however they express no intentional offense. Therefore, this dataset might be more useful for the task of sentiment analysis than for offensive language detection. Table 3 contains examples of texts found in this dataset that are labeled as offensive or deeply offensive but bear no such meaning. Out of the four examples presented in this table only text #2 can be considered offensive.

**Table 2. Arabic offensive language datasets.**

| # | paper | source | size | tags | dialects | focus |
|---|-------|--------|------|------|----------|-------|
| 1 | [13] | Twitter | 6,136 | hate speech, non hate speech | undetermined | religious hate speech |
| 2 | [20] | Twitter | 3,353 | offensive and non-offensive, hate speech, target presence, target | undetermined | – |
| 3 | [25] | Twitter | 5,848 | hate, abusive, normal speech | Levantine | – |
| 4 | [22] | YouTube | 15,050 | offensive, inoffensive | undetermined | – |
| 5 | [17] | Twitter | 10,000 | offensive, vulgar, hate speech, clean | Egyptian, Levantine, Maghrebi, and Gulf | – |
| 6 | [24] | Aljazeera | 31,692 | obscene, offensive, clean | Egyptian | |
| 7 | [24] | Twitter | 1,100 | hate speech, insults, offensive | MSA | – |
| 8 | [18] | social media | 6,024 | abusive, hate, normal | Tunisian (Maghrebi) | – |
| 9 | [23] | Twitter, Facebook, and YouTube | 4,000 | offensive, non-offensive, vulgar, hate speech | various dialects and MSA | – |
| 10 | [27] | YouTube | 8,024 | offensive, non-offensive | Egyptian | public figures |
| 11 | [28] | YouTube | 20,402 | offensive, non-offensive | Moroccan | – |
| 12 | [14] | Twitter | 3,236 | hate speech, not hate speech | undetermined | religious hate speech |
| 13 | [21] | Twitter | 403,688 | severe hate speech, hate speech, ambivalent, no hate speech | undetermined | religious hate speech |
| 14 | [15] | Twitter | 5,240 | None, Discredit, Derailing, Dominance, Stereotyping and Objectification, Threat of violence, Sexual Harassment, and Damning | Levantine | misogyny |
| 15 | [12] | Twitter | 9,196 | offensive, not offensive | MSA and Levantine | – |
| 16 | [19] | Twitter | 23,537 | offensive, not offensive | Saudi | – |

**Table 3. Examples from dataset# 13.**

| # | original text | label | translation |
|---|---|---|---|
| 1 | ط ب وإذاألد أفتقد بالسرطان  ف رح تتعالج؟خل ف قلبٍ بعض من رحمٍ | positive | Ok, and if God hits you with cancer, how will you be treated? Keep some mercy in your heart |
| 2 | اتف مذ عات ,,سخاف ع ولدن ع قل ح ا | positive | The most ridiculous broadcasters and stupidity and childishness and lack of shame |
| 3 | د تورمحد ف مجال لو سمحت تنو متابع لنا | very positive | Dr. Mohamed is in the field. Please follow us |
| 4 | لا طعم للق و  اذا اعدت النفس وشربت ا وح دا | very positive | Coffee has no taste if you make it for yourself and drink it alone |

### 4.3. Dataset statistics

Here we report the basic statistics for the Arabic offensive language datasets. Table 4 shows the minimal, average, and maximal number of words and characters per text in each of the datasets. Additionally, we report the total and average number of non-Arabic characters per dataset; in most cases, these characters are emojis and usernames (including those obtained after anonymization) that appear in social media comments. In one notable case – dataset #11 of [28] – there are texts containing a mixture of French and Arabic or even French texts without Arabic words at all. We have used the NLTK python toolkit [37] to tokenize the texts. We also report actual dataset sizes that might differ from the sizes reported in papers, partially because of empty strings and non-string entries. From here and on, we mark the datasets that address specific dialect(s) that are not MSA by *.

### 4.4. Re-annotation of existing datasets

In this section, we describe how the original labels of datasets that have passed filtering (datasets 2-12 and 14-16) are translated following the new taxonomy. Note that in most cases only the annotation on level 1 (offensive/not offensive) could be extracted; however, some datasets contain annotations that are related to deeper levels of our taxonomy, although not all the levels at once. The translation procedure is outlined in Table 5. In deeper levels of taxonomy with more than two options, such as offense strength, we set the label only when it could be determined. For example, in dataset #5 the label "HS" set to 1 indicates hate speech, but when its value is 0 it is impossible to determine without a new annotation procedure whether the text represents insult, discrediting, or threat.

## 5. Dataset evaluation

### 5.1. Pipeline

The primary stages of our approach are illustrated in Fig 3.

We begin by using re-annotated and filtered datasets, as detailed in Sect 4.1. Next, we analyze each dataset individually, along with a combined MSA and Levantine dataset, which we construct as described in Sect 5.3.3. We assess the performance of each model (outlined in Sect 5.2) with and without fine-tuning on these datasets and present the results.

### 5.2. Pre-trained transformer models

We evaluate several up-to-date pre-trained transformer models for the task of offensive language detection. Note that none of these is trained for detecting offensive language types beyond level 2 of the taxonomy. Four of the models are intended for the Arabic language, and

**Table 4. Arabic offensive language datasets.**

| dataset | min | max | avg | min | max | avg | unique | avg non | size |
|---|---|---|---|---|---|---|---|---|---|
| | cc | cc | cc | wc | wc | wc | wc | Ar cc | |
| #2 | 5 | 140 | 72.02 | 1 | 38 | 14.91 | 15,343 | 7.56 | 3,353 |
| #3* | 3 | 277 | 63.59 | 1 | 52 | 12.09 | 21,514 | 0.29 | 5,848 |
| #4 | 1 | 2,338 | 56.56 | 1 | 447 | 10.67 | 38,280 | 1.00 | 15,050 |
| #5* | 6 | 330 | 102.52 | 1 | 122 | 23.96 | 48,792 | 10.56 | 10,000 |
| #6* | 3 | 200 | 97.79 | 1 | 111 | 18.53 | 103,713 | 0.90 | 31,692 |
| #7 | 3 | 171 | 72.93 | 1 | 32 | 14.31 | 7,289 | 0.84 | 1,100 |
| #8* | 2 | 2,515 | 69.96 | 1 | 388 | 11.74 | 27,171 | 0.00 | 6,024 |
| #9 | 17 | 3,607 | 127.76 | 5 | 655 | 24.35 | 31,903 | 5.82 | 4,000 |
| #10* | 12 | 267 | 69.23 | 2 | 51 | 13.36 | 2,529 | 0.45 | 8,024 |
| #11* | 1 | 899 | 67.74 | 1 | 232 | 13.03 | 50,742 | 12.66 | 20,042 |
| #12 | 20 | 963 | 201.85 | 4 | 159 | 36.41 | 32,622 | 19.65 | 3,236 |
| #14* | 25 | 284 | 121.99 | 3 | 90 | 22.92 | 15,353 | 118.14 | 5,240 |
| #15* | 3 | 330 | 92.50 | 1 | 122 | 21.04 | 43,750 | 8.08 | 9,196 |
| #16* | 2 | 511 | 122.29 | 1 | 121 | 23.73 | 78,739 | 75.70 | 23,537 |

**Table 5. Re-annotating datasets according to the new taxonomy.**

| dataset | original labels | re-annotation |
|---|---|---|
| #2 | sentiment set to "normal" | non-offensive |
| | sentiment other than "normal" | offensive |
| | sentiment includes "abusive" and "hateful" or "disgust" | hate speech |
| | direct label is "direct" | target presence set to "present" |
| | direct label is "indirect" | target presence set to "absent" |
| | target label is "origin" | racism |
| | group label mentions origin | xenophobia |
| | group label is "gay" and the sentiment is not "normal" | homophobia |
| | group label is "gender" and sentiment is not "normal" | sexism |
| #3* | normal | non-offensive |
| | abusive | offensive |
| | hate | offensive, hate speech |
| #4 | N | non-offensive |
| | P | offensive |
| #5*, #8*, #11*, #12, #15*, #16* | 0 | non-offensive |
| | 1 | offensive |
| | HS | offensive, hate speech |
| #6*, #7 | 0 NORMAL_LANGUAGE | non-offensive |
| | -1 OFFENSIVE_LANGUAGE | offensive |
| | -2 OBSCENE_LANGUAGE | offensive and vulgar |
| #9 | Majority_Label is Offensive | offensive |
| | Majority_Label is Non-Offensive | non-offensive |
| #10* | LABEL 0 | non-offensive |
| | LABEL 1 | offensive |
| | in file Politics_Sample and LABEL 1 | ideologism |
| #14* | category is "none" | non-offensive |
| | category other than "none" | offensive |
| | category is "damning" | insult |
| | category is "discredit" | discredit |
| | category is "dominance" or "derailing" | hate speech |
| | misogyny is "misogyny" | sexism |

one is multilingual. Table 6 provides details and references for the models. Below, we also provide some explanation of these models' origins.

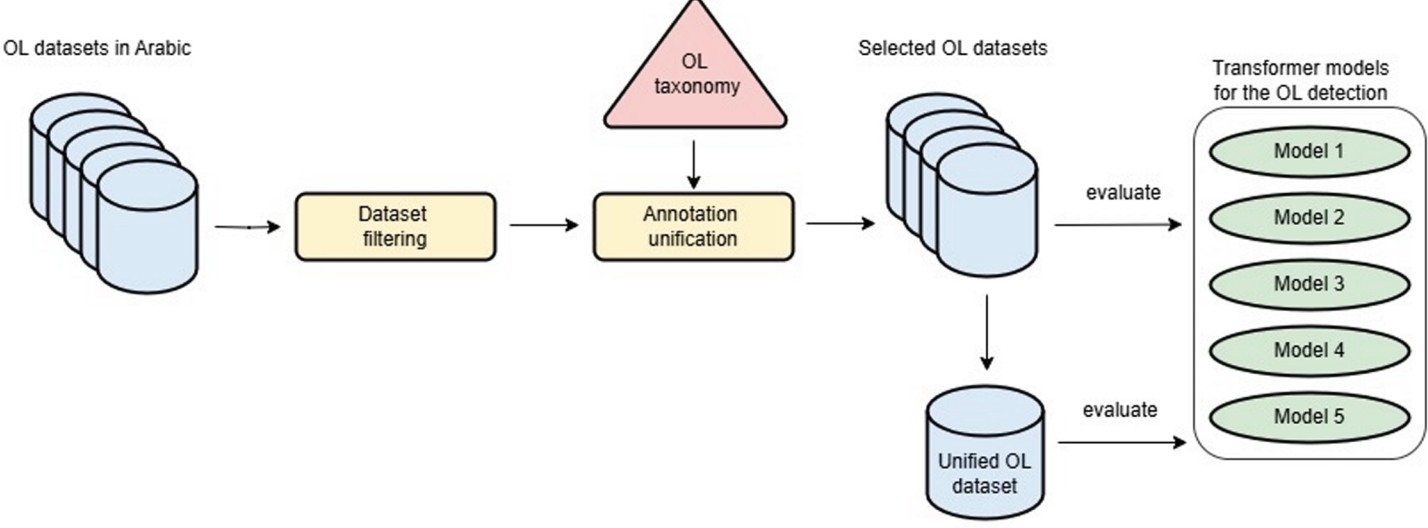

**Fig 3. Evaluation pipeline.**

**Table 6. Arabic and multilingual pre-trained models for offensive language detection.**

| # | huggingface ID | languages | classes |
|---|---|---|---|
| 1 | akhooli/xlm-r-large-arabic-toxic | Arabic | LABEL_0 (not offensive), LABEL_1 (offensive) |
| 2 | Hate-speech-CNERG/dehatebert-mono-arabic | Arabic | HATE (hate speech), NON_HATE (not hate speech) |
| 3 | nourmorsy/PermoBERT-Arabic-Hate-Speech-NoFarasa-WLV-44000Token | Arabic | LABEL_0 (not offensive), LABEL_1 (offensive) |
| 4 | nassga/arabic-offensive-comment-model | Arabic | Offensive, Non-Offensive |
| 5 | christinacdl/XLM_RoBERTa-Multilingual-Hate-Speech-Detection-New | multilingual | HATEFUL, NOT |

The XLM-RoBERTa [38] model, a RoBERTa model variation trained by Facebook AI, serves as the foundation for Model 1 [39]. It has been specifically tailored to identify harmful words or material in Arabic text. It can successfully identify instances of toxicity in text data and makes use of transfer learning techniques to comprehend the subtleties of toxic language in Arabic.

The purpose of Model 2 [40] is to recognize hate speech in Arabic texts. In order to capture the distinct language patterns and settings linked to hate speech, it is refined using data that includes examples of hate speech in Arabic.

Model 3 [41] is based on BERT architecture and it is fine-tuned using hate speech datasets in Arabic, but it does not utilize the Farasa segmentation tool during training. Vocabulary size of this model is 44,000.

The purpose of Model 4 [42] is to recognize words or content that might be deemed offensive or unsuitable in a certain situation. It analyzes and categorizes text data using methods from machine learning and natural language processing.

The multilingual hate speech detection model 5 [43] is trained with the XLM-RoBERTa model, which has been pre-trained on 2.5TB of data comprising 100 languages. By utilizing

XLM-RoBERTa's multilingual capabilities, it seeks to recognize and categorize hate speech in a variety of languages and comprehend its contexts and patterns.

## 5.3. Evaluation of datasets

In this section, we present the evaluation results of pre-trained models for offensive language detection described. We used level 1 annotation of texts (offensive or not offensive) that is available for all the datasets described in Sect 4. The majority rule and labels at this level of taxonomy are shown in Table 7.

Because of text length and number of token limitations, we have omitted all the texts with more than 512 tokens or more than 1,000 characters. We randomly split every dataset into 80% train and 20% test sets.

**5.3.1. Without fine-tuning.** In this part of the evaluation procedure, we used pre-trained models as is and applied them to the test sets of every dataset. Tables 8–12 report the results for each case, including precision, recall, accuracy, and the F1 measure. The cases where a model performed, in terms of accuracy, better than a majority rule shown in Table 7, are marked with gray.

Fig 4 shows the accuracy of each model across all datasets and summarizes the results.

Table 13 shows the best accuracy and the models producing them for all datasets. We can see that in most cases the best result is above the majority, but not for datasets 6 and 12. Dataset 6 is derived from comments on news articles, and dataset 12 is dedicated to religious hate speech that seems to be harder to detect. Model 4 trained in Arabic supplies the best scores in most cases, but a multilingual model 5 performs well too. Model 5 produces good

**Table 7. Majority rule and classes for all datasets at the level 1 of taxonomy.**

| dataset | maj | class | dataset | maj | class |
|---------|--------|-------|---------|--------|-------|
| #2 | 0.7265 | 1 | #9 | 0.8304 | 0 |
| #3* | 0.6239 | 0 | #10* | 0.5380 | 1 |
| #4 | 0.6139 | 0 | #11* | 0.6218 | 0 |
| #5* | 0.8009 | 0 | #12 | 0.8006 | 0 |
| #6* | 0.8216 | 1 | #14* | 0.5067 | 0 |
| #7 | 0.5864 | 1 | #15* | 0.6128 | 0 |
| #8* | 0.6337 | 0 | #16* | 0.5259 | 0 |

**Table 8. Evaluation of pre-trained model #1 for offensive language detection at level 1.**

| model | data | P | R | F1 | acc |
|-------|------|--------|--------|--------|--------|
| 1 | #2 | 0.6448 | 0.6502 | 0.6330 | 0.6502 |
| 1 | #3* | 0.7591 | 0.7607 | 0.7597 | 0.7607 |
| 1 | #4 | 0.7332 | 0.7310 | 0.7278 | 0.7310 |
| 1 | #5* | 0.7855 | 0.7944 | 0.7812 | 0.7944 |
| 1 | #6* | 0.6866 | 0.6037 | 0.5583 | 0.6037 |
| 1 | #7 | 0.7129 | 0.7000 | 0.6979 | 0.7000 |
| 1 | #8* | 0.7067 | 0.6611 | 0.6564 | 0.6611 |
| 1 | #9 | 0.7617 | 0.7688 | 0.7459 | 0.7688 |
| 1 | #10* | 0.6693 | 0.6409 | 0.6437 | 0.6409 |
| 1 | #11* | 0.6568 | 0.6327 | 0.6418 | 0.6327 |
| 1 | #12 | 0.7276 | 0.4606 | 0.4321 | 0.4606 |
| 1 | #14* | 0.7537 | 0.7261 | 0.7303 | 0.7261 |
| 1 | #15* | 0.7866 | 0.7874 | 0.7869 | 0.7874 |
| 1 | #16* | 0.7690 | 0.7090 | 0.7200 | 0.7090 |

**Table 9. Evaluation of pre-trained model #2 for offensive language detection at level 1.**

| model | data | P | R | F1 | acc |
|---|---|---|---|---|---|
| 2 | #2 | 0.7313 | 0.6786 | 0.6613 | 0.6786 |
| 2 | #3* | 0.8968 | 0.7410 | 0.7809 | 0.7410 |
| 2 | #4 | 0.8453 | 0.6787 | 0.7273 | 0.6787 |
| 2 | #5* | 0.6595 | 0.6563 | 0.6229 | 0.6563 |
| 2 | #6* | 0.8585 | 0.2811 | 0.3115 | 0.2811 |
| 2 | #7 | 0.8212 | 0.5182 | 0.5742 | 0.5182 |
| 2 | #8* | 0.8210 | 0.6902 | 0.7299 | 0.6902 |
| 2 | #9 | 0.9023 | 0.8643 | 0.8786 | 0.8643 |
| 2 | #10* | 0.8573 | 0.4903 | 0.5883 | 0.4903 |
| 2 | #11* | 0.8178 | 0.6092 | 0.6823 | 0.6092 |
| 2 | #12 | 0.5810 | 0.5981 | 0.5589 | 0.5981 |
| 2 | #14* | 0.8072 | 0.5868 | 0.6396 | 0.5868 |
| 2 | #15* | 0.5794 | 0.5780 | 0.5787 | 0.5780 |
| 2 | #16* | 0.8643 | 0.6134 | 0.6758 | 0.6134 |

**Table 10. Evaluation of pre-trained model #3 for offensive language detection at level 1.**

| model | data | P | R | F1 | acc |
|---|---|---|---|---|---|
| 3 | #2 | 0.8380 | 0.6801 | 0.7457 | 0.6801 |
| 3 | #3* | 0.7515 | 0.3709 | 0.4599 | 0.3709 |
| 3 | #4 | 0.5881 | 0.3987 | 0.4329 | 0.3987 |
| 3 | #5* | 0.7639 | 0.2886 | 0.3152 | 0.2886 |
| 3 | #6* | 0.8438 | 0.7654 | 0.8009 | 0.7654 |
| 3 | #7 | 0.8224 | 0.5864 | 0.6642 | 0.5864 |
| 3 | #8* | 0.5166 | 0.4352 | 0.4379 | 0.4352 |
| 3 | #9 | 0.7810 | 0.2374 | 0.2685 | 0.2374 |
| 3 | #10* | 0.3487 | 0.3281 | 0.3234 | 0.3281 |
| 3 | #11* | 0.3487 | 0.3281 | 0.3234 | 0.3281 |
| 3 | #12 | 0.8211 | 0.2566 | 0.3105 | 0.2566 |
| 3 | #14* | 0.7074 | 0.4876 | 0.5491 | 0.4876 |
| 3 | #15* | 0.7468 | 0.4203 | 0.4885 | 0.4203 |
| 3 | #16* | 0.6141 | 0.6124 | 0.6130 | 0.6124 |

**Table 11. Evaluation of pre-trained model #4 for offensive language detection at level 1.**

| model | data | P | R | F1 | acc |
|---|---|---|---|---|---|
| 4 | #2 | 0.7906 | 0.7309 | 0.7537 | 0.7309 |
| 4 | #3* | 0.8299 | 0.8308 | 0.8302 | 0.8308 |
| 4 | #4 | 0.8051 | 0.7925 | 0.7962 | 0.7925 |
| 4 | #5* | 0.7648 | 0.4912 | 0.4622 | 0.4912 |
| 4 | #6* | 0.7673 | 0.4442 | 0.4116 | 0.4442 |
| 4 | #7 | 0.7944 | 0.7182 | 0.7210 | 0.7182 |
| 4 | #8* | 0.7632 | 0.7633 | 0.7632 | 0.7633 |
| 4 | #9 | 0.7859 | 0.6947 | 0.6567 | 0.6947 |
| 4 | #10* | 0.8254 | 0.5719 | 0.6234 | 0.5719 |
| 4 | #11* | 0.8962 | 0.6491 | 0.7308 | 0.6491 |
| 4 | #12 | 0.7652 | 0.3364 | 0.3435 | 0.3364 |
| 4 | #14* | 0.9788 | 0.5115 | 0.6565 | 0.5115 |
| 4 | #15* | 0.7045 | 0.5644 | 0.5762 | 0.5644 |
| 4 | #16* | 0.8398 | 0.7409 | 0.7570 | 0.7409 |

**Table 12. Evaluation of pre-trained model #5 for offensive language detection at level 1.**

| model | data | P | R | F1 | acc |
|---|---|---|---|---|---|
| 5 | #2 | 0.6984 | 0.5172 | 0.5068 | 0.5172 |
| 5 | #3* | 0.7518 | 0.7205 | 0.7303 | 0.7205 |
| 5 | #4 | 0.7352 | 0.6940 | 0.7067 | 0.6940 |
| 5 | #5* | 0.8807 | 0.8489 | 0.8607 | 0.8489 |
| 5 | #6* | 0.7204 | 0.4206 | 0.3870 | 0.4206 |
| 5 | #7 | 0.7469 | 0.6455 | 0.6529 | 0.6455 |
| 5 | #8* | 0.6511 | 0.6473 | 0.6490 | 0.6473 |
| 5 | #9 | 0.8840 | 0.8360 | 0.8555 | 0.8360 |
| 5 | #10* | 0.6838 | 0.5976 | 0.6111 | 0.5976 |
| 5 | #11* | 0.7317 | 0.6418 | 0.6712 | 0.6418 |
| 5 | #12 | 0.6163 | 0.6430 | 0.6115 | 0.6430 |
| 5 | #14* | 0.8065 | 0.6832 | 0.7049 | 0.6832 |
| 5 | #15* | 0.8309 | 0.7645 | 0.7804 | 0.7645 |
| 5 | #16* | 0.7879 | 0.6308 | 0.6667 | 0.6308 |

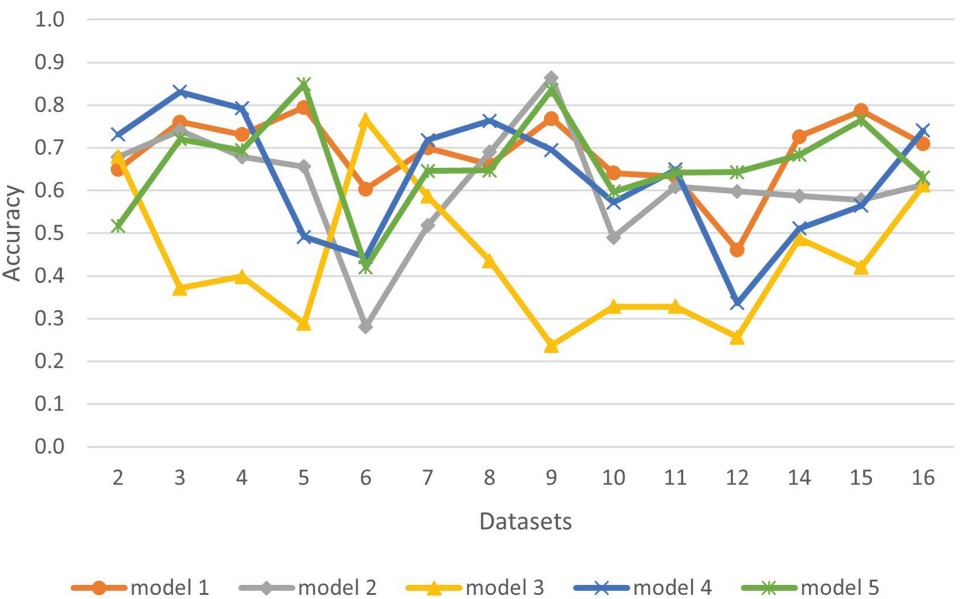

**Fig 4. Comparative performance of transformer models without fine tuning.**

results for all but three datasets (2,6,12). The hardest datasets are 6 and 12, where no pre-trained model succeeded in achieving accuracy above the majority. Dataset 16 seems to be the easiest because all pre-trained models perform well on it.

**5.3.2. With fine-tuning.** In this part of the evaluation, we fine-tune pre-trained models on the training sets and evaluate them on the test sets of every dataset, with the train/test split being 80%/20%. We have fine-tuned every model for 3 epochs with batch size 16, Adam optimizer, and standard learning rate of 0.00002. All texts were padded to the maximal length, and the attention mask was set to ignore the padded tokens.

Tables 14–18 report the results for all models. The cases where a model performed, in terms of accuracy, better than a majority rule (detailed in Table 7), are marked with gray color.

**Table 13. Best scores and models for all datasets, without fine tuning.**

| dataset | max acc | the best model | above majority | pre-trained models above majority |
|---|---|---|---|---|
| #2 | 0.7537 | 4 | yes | 4 |
| #3* | 0.8302 | 4 | yes | 1,2,4,5 |
| #4 | 0.7962 | 4 | yes | 1,2,4,5 |
| #5* | 0.8489 | 5 | yes | 1,5 |
| #6* | 0.7654 | 3 | no | – |
| #7 | 0.7210 | 4 | yes | 1,4,5 |
| #8* | 0.7632 | 4 | yes | 1,2,4,5 |
| #9 | 0.8643 | 2 | yes | 2,5 |
| #10* | 0.6409 | 1 | yes | 4,5 |
| #11* | 0.7308 | 4 | yes | 1,4,5 |
| #12 | 0.6430 | 5 | no | – |
| #14* | 0.7261 | 1 | yes | 1,2,4,5 |
| #15* | 0.7874 | 1 | yes | 1,5 |
| #16* | 0.7570 | 4 | yes | 1,2,3,4,5 |

**Table 14. Evaluation of fine-tuned model #1 for offensive language detection at level 1.**

| model | data | P | R | F1 | acc |
|---|---|---|---|---|---|
| 1 | #2 | 0.5976 | 0.6138 | 0.5993 | 0.6517 |
| 1 | #3* | 0.7651 | 0.7759 | 0.7685 | 0.7778 |
| 1 | #4 | 0.7246 | 0.7367 | 0.7228 | 0.7272 |
| 1 | #5* | 0.6861 | 0.7623 | 0.6993 | 0.7629 |
| 1 | #6* | 0.6058 | 0.6805 | 0.5610 | 0.6069 |
| 1 | #7 | 0.7508 | 0.7564 | 0.7405 | 0.7409 |
| 1 | #8* | 0.6442 | 0.6538 | 0.6265 | 0.6290 |
| 1 | #9 | 0.6753 | 0.7753 | 0.6901 | 0.7688 |
| 1 | #10* | 0.6579 | 0.6561 | 0.6499 | 0.6502 |
| 1 | #11* | 0.5937 | 0.5867 | 0.5879 | 0.6253 |
| 1 | #12 | 0.5757 | 0.5828 | 0.3875 | 0.3879 |
| 1 | #14* | 0.7136 | 0.7111 | 0.7108 | 0.7118 |
| 1 | #15* | 0.7460 | 0.7567 | 0.7481 | 0.7548 |
| 1 | #16* | 0.4727 | 0.4996 | 0.3470 | 0.5239 |

**Table 15. Evaluation of fine-tuned model #2 for offensive language detection at level 1.**

| model | data | P | R | F1 | acc |
|---|---|---|---|---|---|
| 2 | #2 | 0.1358 | 0.5000 | 0.2136 | 0.2716 |
| 2 | #3* | 0.1878 | 0.5000 | 0.2731 | 0.3757 |
| 2 | #4 | 0.5136 | 0.5073 | 0.4732 | 0.5864 |
| 2 | #5* | 0.6000 | 0.5018 | 0.1702 | 0.2023 |
| 2 | #6* | 0.6038 | 0.5046 | 0.4641 | 0.8202 |
| 2 | #7 | 0.4865 | 0.4861 | 0.4846 | 0.4926 |
| 2 | #8* | 0.1832 | 0.5000 | 0.2681 | 0.3664 |
| 2 | #9 | 0.5248 | 0.5438 | 0.4623 | 0.5161 |
| 2 | #10* | 0.2306 | 0.4989 | 0.3154 | 0.4607 |
| 2 | #11* | 0.5225 | 0.5003 | 0.2774 | 0.3791 |
| 2 | #12 | 0.1000 | 0.5000 | 0.1667 | 0.2000 |
| 2 | #14* | 0.5129 | 0.5007 | 0.3438 | 0.4969 |
| 2 | #15* | 0.4535 | 0.4656 | 0.4466 | 0.5288 |
| 2 | #16* | 0.2376 | 0.5000 | 0.3221 | 0.4751 |

**Table 16. Evaluation of fine-tuned model #3 for offensive language detection at level 1.**

| model | data | P | R | F1 | acc |
|---|---|---|---|---|---|
| 3 | #2 | 0.3632 | 0.5000 | 0.4208 | 0.7265 |
| 3 | #3* | 0.1880 | 0.5000 | 0.2733 | 0.3761 |
| 3 | #4 | 0.3893 | 0.4963 | 0.3814 | 0.6083 |
| 3 | #5* | 0.4005 | 0.5000 | 0.4447 | 0.8009 |
| 3 | #6* | 0.4108 | 0.5000 | 0.4510 | 0.8216 |
| 3 | #7 | 0.2932 | 0.5000 | 0.3696 | 0.5864 |
| 3 | #8* | 0.8173 | 0.5011 | 0.3905 | 0.6349 |
| 3 | #9 | 0.4156 | 0.5000 | 0.4539 | 0.8313 |
| 3 | #10* | 0.4827 | 0.4893 | 0.4229 | 0.4658 |
| 3 | #11* | 0.1892 | 0.5000 | 0.2745 | 0.3783 |
| 3 | #12 | 0.4003 | 0.5000 | 0.4446 | 0.8006 |
| 3 | #14* | 0.2533 | 0.5000 | 0.3363 | 0.5067 |
| 3 | #15* | 0.1936 | 0.5000 | 0.2791 | 0.3872 |
| 3 | #16* | 0.2377 | 0.5000 | 0.3222 | 0.4755 |

**Table 17. Evaluation of fine-tuned model #4 for offensive language detection at level 1.**

| model | data | P | R | F1 | acc |
|---|---|---|---|---|---|
| 4 | #2 | 0.1368 | 0.5 | 0.2148 | 0.2735 |
| 4 | #3* | 0.188 | 0.5 | 0.2733 | 0.3761 |
| 4 | #4 | 0.3068 | 0.5 | 0.3803 | 0.6136 |
| 4 | #5* | 0.4718 | 0.4994 | 0.4464 | 0.7984 |
| 4 | #6* | 0.3392 | 0.4997 | 0.1515 | 0.1784 |
| 4 | #7 | 0.2932 | 0.5 | 0.3696 | 0.5864 |
| 4 | #8* | 0.317 | 0.5 | 0.3880 | 0.634 |
| 4 | #9 | 0.5122 | 0.5049 | 0.2018 | 0.2112 |
| 4 | #10* | 0.4607 | 0.4865 | 0.3705 | 0.4558 |
| 4 | #11* | 0.4391 | 0.4999 | 0.2748 | 0.3783 |
| 4 | #12 | 0.3998 | 0.4971 | 0.4432 | 0.796 |
| 4 | #14* | 0.2467 | 0.5 | 0.3304 | 0.4933 |
| 4 | #15* | 0.4871 | 0.4872 | 0.4605 | 0.4606 |
| 4 | #16* | 0.2377 | 0.5 | 0.3222 | 0.4755 |

**Table 18. Evaluation of fine-tuned model #5 for offensive language detection at level 1.**

| model | data | P | R | F1 | acc |
|---|---|---|---|---|---|
| 5 | #2 | 0.6002 | 0.6180 | 0.5319 | 0.5366 |
| 5 | #3* | 0.7173 | 0.6996 | 0.7051 | 0.7333 |
| 5 | #4 | 0.6789 | 0.6538 | 0.6577 | 0.6947 |
| 5 | #5* | 0.7215 | 0.7028 | 0.7112 | 0.8234 |
| 5 | #6* | 0.5820 | 0.6161 | 0.4280 | 0.4357 |
| 5 | #7 | 0.7248 | 0.6980 | 0.6553 | 0.6591 |
| 5 | #8* | 0.6476 | 0.6460 | 0.6467 | 0.6739 |
| 5 | #9 | 0.6511 | 0.6297 | 0.6385 | 0.8113 |
| 5 | #10* | 0.6366 | 0.6212 | 0.6004 | 0.6075 |
| 5 | #11* | 0.6007 | 0.5749 | 0.5698 | 0.6371 |
| 5 | #12 | 0.5974 | 0.6500 | 0.5787 | 0.6352 |
| 5 | #14* | 0.7055 | 0.6670 | 0.6525 | 0.6698 |
| 5 | #15* | 0.7782 | 0.7298 | 0.7395 | 0.7705 |
| 5 | #16* | 0.4799 | 0.4951 | 0.3870 | 0.5165 |

Fig 5 shows the accuracy of each model across all datasets and summarizes the results.

Table 19 shows the best accuracy and the models producing them for all datasets. The arrows indicate cases where an improvement was achieved in comparison with the best accuracy achieved by models that were not fine-tuned. It is interesting to see that only four datasets (6,7,10, and 12) improvement was achieved, showing that variations in the training and tuning data do not necessarily increase classification accuracy.

We also report the models that produced results better than the majority and the models that outperformed the best pre-trained models. We observe for datasets 2,6,12, and 16 no model achieved accuracy above the majority. For the rest of the datasets, the best models are 1 and 5 because they produce accuracy above the majority for eight datasets. However, in terms of the best accuracy, model 1 wins in 7 cases model 5 – in 4 cases, and model 3 in 4 cases. Models 2 and 4 underperform, with model 2 producing the worst results (see Table 15). It is interesting to see that model 3 produced the best results for all but one dataset in cases where the majority is not outperformed.

Based on the results of fine-tuning for Arabic offensive language detection presented in Table 19, we suggest the following recommendations for enhancing model performance when targeting specific Arabic dialects.

- To incorporate preprocessing techniques tailored to specific dialects, such as normalizing dialectal variations and unique morphological patterns.
- To augment datasets with synthetic examples that reflect diverse dialectal features can help mitigate data scarcity issues.
- To apply dialect-specific transfer learning by fine-tuning models on datasets that are exclusively representative of a target dialect before applying them to mixed or broader datasets could lead to more effective feature learning.

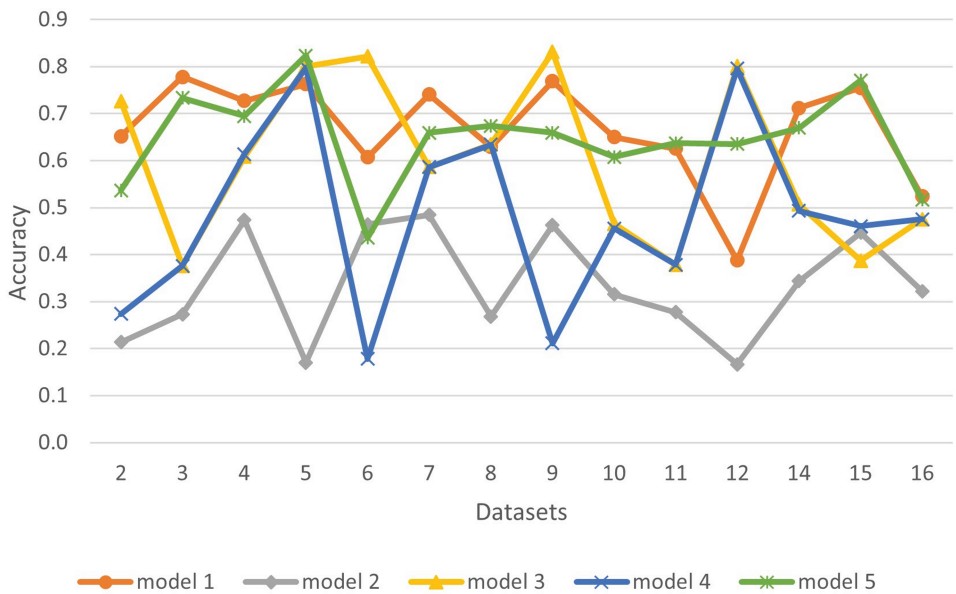

**Fig 5. Comparative performance of transformer models with fine tuning.**

**Table 19. Best scores and models for all datasets, with fine tuning.**

| dataset | max acc | the best model | above majority | pre-trained models above majority | above the best pre-trained model |
|---------|---------|----------------|----------------|-----------------------------------|----------------------------------|
| #2 | 0.7265 | 3 | no | – | no |
| #3* | 0.7778 | 1 | yes | 1,5 | no |
| #4 | 0.7272 | 1 | yes | 1,5 | no |
| #5* | 0.8234 | 5 | yes | 5 | no |
| #6* | 0.8216↑ | 3 | no | – | yes |
| #7 | 0.7409↑ | 1 | yes | 1,5 | yes |
| #8* | 0.6739 | 5 | yes | 3,4,5 | no |
| #9 | 0.8313 | 3 | yes | 3 | no |
| #10* | 0.6502↑ | 1 | yes↑ | 1,5 | yes |
| #11* | 0.6371 | 5 | yes | 1,5 | no |
| #12 | 0.8006↑ | 3 | no | – | yes |
| #14* | 0.7118 | 1 | yes | 1,5 | no |
| #15* | 0.7705 | 5 | yes | 1,5 | no |
| #16* | 0.5239 | 1 | no | – | no |

**Table 20. Joint MSA dataset evaluation.**

| model | P | R | F1 | acc |
|-------|---|---|----|----|
| pre-trained | | | | |
| 1 | 0.7075 | 0.6976 | 0.6934 | 0.6976 |
| 4 | 0.8121 | 0.8024 | 0.8053 | 0.8024 |
| 5 | 0.7191 | 0.6944 | 0.7026 | 0.6944 |
| fine-tuned | | | | |
| 1 | 0.6817 | 0.6920 | 0.6796 | 0.6852 |
| 4 | 0.8084 | 0.5005 | 0.3824 | 0.6169 |
| 5 | 0.6737 | 0.6467 | 0.6504 | 0.6910 |

**5.3.3. Joint MSA data evaluation.** To evaluate the datasets further, we unified datasets that are written in MSA (datasets 2,4,7,9,12) and excluded those that contain a specific dialect or mix of dialects. The resulting dataset has 26,739 texts, with majority 0.6165 at level 1 and the majority class 0 or "non-offensive". We have evaluated the best-performing pre-trained models (1,4, and 5) with and without fine-tuning on this dataset with the random 80%/20% train/test split. Evaluation results appear in Table 20. We see that fine-tuning did not improve classification accuracy, which could be an indication of poor generalization of a model or the differences in training and tuning data. The model can forget what it learned during pre-training if the fine-tuning dataset is very different from the pre-training data.

# 6. Conclusions and future work

This work presented a novel, structured taxonomy for Arabic offensive language detection. This taxonomy offers a foundation for consistent data annotation and facilitates the development of more robust detection models. We analyzed 16 existing datasets to assess their alignment with the taxonomy and compatibility across dialects. Our analysis identified potential gaps and biases in existing datasets, highlighting areas for future data collection efforts. Additionally, we explored the transferability of deep learning models, particularly pre-trained and fine-tuned transformers, across datasets from different dialects.

Our findings hold significant implications for future research in Arabic offensive language detection. The proposed taxonomy can guide the creation of new, high-quality datasets that

comprehensively cover the spectrum of Arabic offensive language. Future research can leverage this taxonomy to develop more balanced and culturally aware datasets, ensuring models are trained on a wider range of offensive language types.

The analysis of existing Arabic offensive language datasets reveals that they cover various dialects, topics, and social media platforms, with Twitter being the most frequently used source. Despite the variety, two datasets were excluded from further examination due to issues with data retrieval and label suitability. The remaining datasets provide valuable resources for studying and detecting offensive language in Arabic, though re-annotation was necessary to align with a new taxonomy for more consistent classification. As part of our future work, we aim to explore the use of mixed-dialect datasets, despite their lack of dialect annotations, to enhance the robustness of models for offensive language detection across a broader range of Arabic dialects.

Without fine-tuning, pre-trained models generally achieved better accuracy than a majority rule baseline, demonstrating their effectiveness in offensive language detection across various datasets. Model 4, trained in Arabic, consistently provided the best accuracy, while multilingual Model 5 performed well across most datasets. However, datasets 6 (news comments) and 12 (religious hate speech) were particularly challenging, with no model surpassing the majority baseline accuracy. Therefore, dialect-specific models may be necessary for optimal performance. Future work can explore the development of models tailored to specific dialects, considering the unique cultural nuances and linguistic variations within each dialect.

Fine-tuning improved the performance of certain models. However, fine-tuning did not guarantee improvement for all datasets, highlighting the variability and dataset dependency in model performance. We have also compiled a unified MSA dataset and showed that fine-tuning does not improve the classification accuracy, indicating possible poor generalization or discrepancies between pre-training and fine-tuning data.

Offensive language detection is crucial for online communication across all languages. Future research can explore the adaptation of the proposed taxonomy or its core principles for offensive language detection in other languages, fostering a more inclusive online environment globally. It will also focus on detecting implicit offensive language in Arabic, which poses a bigger challenge due to the subtlety and context-dependency of such expressions. Moreover, the lack of implicit offensive language datasets for Arabic poses an additional challenge.

## 7. Limitations

Dataset analysis limitations include the exclusion of two datasets due to retrieval issues and misalignment with offensive language detection, potentially limiting the comprehensiveness of the analysis. Additionally, the manual re-annotation process might introduce subjective biases and inconsistencies. Finally, focusing on specific social media platforms like Twitter may not fully capture the diversity of offensive language across different online contexts.

We notice that relying on pre-trained models without fine-tuning for offensive language detection in Arabic may not fully capture the specific nuances of each dataset. The performance disparity across datasets, especially the poor results for datasets 6 and 12, highlights potential limitations in the generalizability of pre-trained models to different types of offensive language. The fine-tuning process did not consistently improve model performance for all datasets, with some datasets showing no improvement over the majority rule. Our results are limited to the chosen datasets, and the findings may not generalize to other types of data or different domains. The findings of the unified MSA dataset evaluation suggest that the fine-tuning process can lead to decreased performance if the fine-tuning data significantly

differs from the pre-training data, highlighting the need to develop new and robust offensive language detection models for Arabic.

## Supporting information

**S1 Appendix**
(PDF)

## Acknowledgments

The authors express their appreciation to Ali El-Afaoui, Muhammad Abu Jaafar, and Samer Masaed for their valuable help with data analysis and annotation. This work was supported by the Israel Innovation Authority. The subject of the program is the development of a dataset and a language model for identifying offensive language in Hebrew and Arabic.

## Author contributions

**Conceptualization:** Natalia Vanetik.

**Data curation:** Yossef Haim Shrem.

**Formal analysis:** Natalia Vanetik.

**Funding acquisition:** Chaya Liebeskind.

**Investigation:** Chaya Liebeskind, Yossef Haim Shrem, Marina Litvak, Natalia Vanetik.

**Methodology:** Chaya Liebeskind, Marina Litvak.

**Writing – original draft:** Chaya Liebeskind, Yossef Haim Shrem, Marina Litvak, Natalia Vanetik.

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
