## [Decision Letter · Decision Letter 0]

4 Dec 2024

PONE-D-24-33044Unveiling the Spectrum of Arabic Offensive Language: Taxonomy and InsightsPLOS ONE

Dear Dr. Liebeskind,

Thank you for submitting your manuscript to PLOS ONE. After careful consideration, we feel that it has merit but does not fully meet PLOS ONE’s publication criteria as it currently stands. Therefore, we invite you to submit a revised version of the manuscript that addresses the points raised during the review process.

We look forward to receiving your revised manuscript.

Kind regards,

Hikmat Ullah Khan, PhD (Computer Science)

Academic Editor

PLOS ONE

Journal Requirements:

2. In your Methods section, please include additional information about your dataset and ensure that you have included a statement specifying whether the collection and analysis method complied with the terms and conditions for the source of the data.

3. Your abstract cannot contain citations. Please only include citations in the body text of the manuscript, and ensure that they remain in ascending numerical order on first mention.

Additional Editor Comments :

The overall research work is good and the authors are required to address the reviewers' comments. The authors have added table for comparison of datasets and the evaluation of results. A table of comparison of literature review is also expected sharing the details, such as Ref no, Year of Research work to show the chronological order of the work, methods used, features used for analysis, name of dataset and results. Thank you

Reviewers' comments:

Reviewer's Responses to Questions

**Comments to the Author**

1. Is the manuscript technically sound, and do the data support the conclusions?

Reviewer #1: Yes

Reviewer #2: Yes

2. Has the statistical analysis been performed appropriately and rigorously? 

Reviewer #1: Yes

Reviewer #2: Yes

3. Have the authors made all data underlying the findings in their manuscript fully available?

Reviewer #1: Yes

Reviewer #2: Yes

4. Is the manuscript presented in an intelligible fashion and written in standard English?

Reviewer #1: Yes

Reviewer #2: Yes

5. Review Comments to the Author

Reviewer #1: The manuscript you provided, titled Unveiling the Spectrum of Arabic Offensive Language: Taxonomy and Insights, presents a novel taxonomy to classify offensive language in Arabic, contributing significantly to a domain primarily dominated by research on Indo-European languages. It introduces seven distinct levels of offensive language and tests various pre-trained and fine-tuned transformer models for detecting such language across multiple Arabic dialects.

Peer Review for PLOS ONE:

1. Overall Summary and Contribution

This manuscript introduces a novel taxonomy for categorizing Arabic offensive language, filling a gap in offensive language detection research focused primarily on Indo-European languages. The taxonomy is applied to existing datasets, and transformer models are tested to evaluate offensive language detection performance. This paper is timely and contributes significantly to Arabic natural language processing (NLP), sociocultural studies, and the development of machine learning models for offensive language detection in Arabic.

2. Originality

The work is highly original, as it introduces a taxonomy specifically for Arabic offensive language, a less studied area in NLP. The paper’s focus on cultural and linguistic diversity provides new insights into how offensive language is structured in Semitic languages, making it valuable to multiple fields.

3. Strengths

• Novel Taxonomy: The manuscript presents a well-structured taxonomy that can serve as a foundational framework for future studies in offensive language detection in Semitic languages.

• Data Analysis: A thorough analysis of existing Arabic datasets and their compatibility with the new taxonomy is performed. This highlights gaps in current data and sets the stage for future data collection efforts.

• Transformers and Fine-Tuning: The authors provide a well-executed evaluation of the performance of pre-trained and fine-tuned transformer models on Arabic datasets, demonstrating the importance of linguistic and dialectal nuances.

• Cultural Sensitivity: The paper acknowledges the challenges posed by the cultural and dialectal diversity of Arabic, which enriches the discussion and improves the applicability of the taxonomy.

4. Weaknesses

• Limited Exploration of Implicit Language: While the paper briefly introduces implicit offensive language (level 7), it does not delve deeply into this area, which could be an interesting direction for future research. The exclusion of implicit language due to its complexity is understandable but leaves a gap in addressing the subtler aspects of offensive communication.

• Dataset Quality: Although the analysis of existing datasets is commendable, the authors could have offered more insight into the annotation processes of the datasets and the consistency of labeling across different datasets. Future work could consider developing a more comprehensive and standardized Arabic offensive language dataset.

• Model Fine-Tuning: The performance of the transformer models varies across datasets, and more discussion on fine-tuning strategies specific to Arabic dialects could enhance the impact of the findings. The paper could benefit from a more in-depth explanation of why certain models underperformed in specific cases and how dialectal differences influenced these results.

5. Clarity and Presentation

• The manuscript is generally clear and well-structured. The sections logically flow from one to the next, and the discussion is backed by solid reasoning.

• Figures and Tables: The tables summarizing dataset details and model performance are useful, but the clarity of the figures could be improved. Additionally, the discussion surrounding Table 8 could be expanded to offer more critical insights into model performance variations.

6. Recommendations

• Expand on Implicit Language: Future research should aim to explore implicit offensive language more thoroughly. As this is a significant area in offensive communication, addressing it would provide a more complete understanding of the phenomenon.

• Dataset Unification: Consider developing a unified, annotated dataset that covers the wide spectrum of Arabic dialects, as this would improve the generalizability of model performance.

• Fine-Tuning Insights: Offering more detailed recommendations on fine-tuning models for specific Arabic dialects would improve practical applications for offensive language detection systems.

7. Conclusion

Overall, the manuscript is a valuable contribution to Arabic NLP, providing a detailed taxonomy that future researchers can build upon. With some refinements, particularly in the treatment of implicit offensive language and dataset standardization, the work will have a lasting impact on the field.

Recommendation: Minor Revisions

This paper is well-positioned for publication but would benefit from addressing the points raised regarding implicit language, dataset quality, and model fine-tuning strategies.

Reviewer #2: The proposed study focuses on classifying offensive language in Arabic and introduces a taxonomy that categorizes offensive language into seven distinct levels. The authors evaluate the performance of pre-trained and fine-tuned Arabic transformer models for offensive language detection using five datasets. The overall manuscript is well-written; however, the following suggestions can further enhance its quality:

Applications and Advantages: The authors should include a paragraph in the introduction highlighting the practical applications and benefits of the proposed work. This will help contextualize its significance.

Improved Contributions Section: The contribution paragraph should be revised to explicitly specify the techniques or added features that contribute to the improved accuracy of the proposed models.

Enhanced Related Work Section: The related work section can be strengthened by connecting studies based on publication year or thematic similarities. Rather than simply referencing studies as "the paper [23]" or "the study in [42]," the authors could cite them using author names (e.g., "Levi et al.") or mention the models or approaches discussed in those studies.

Pictorial Representation: Including a pictorial representation of the study's steps or framework would significantly improve readers' understanding of the proposed methodology.

Results Presentation: The results section can be improved by incorporating a mix of graphical and tabular representations. Graphs can provide a visual comparison, while tables, as seen in Tables 9-19, can offer detailed numerical insights, facilitating a comprehensive performance analysis against baseline models.

Future Work: Authors can improve the future work with more concrete research directions.

6. PLOS authors have the option to publish the peer review history of their article (what does this mean?). If published, this will include your full peer review and any attached files.

Reviewer #1: **Yes: **Mohammed Alghobiri

Reviewer #2: No

---

## [Author Response · Author response to Decision Letter 1]

2 Jan 2025

Response to reviewers

We are very grateful to the reviewers and the academic editor for their comments and suggestions on our manuscript, PONE-D-24-33044, titled “Unveiling the Spectrum of Arabic Offensive Language: Taxonomy and Insights,” which have allowed us to improve the quality of our work. We describe below how we addressed all the issues reported by the editor and reviewers. All the changes are marked in the updated manuscript in blue font color.

Editorial comments

COMMENT 1: Please ensure that your manuscript meets PLOS ONE's style requirements, including those for file naming. The PLOS ONE style templates can be found at

RESPONSE 1: We have ensured that we use PLOS ONE latex template to format our manuscript.

COMMENT 2: In your Methods section, please include additional information about your dataset and ensure that you have included a statement specifying whether the collection and analysis method complied with the terms and conditions for the source of the data.

RESPONSE 2: We have added the following statement to the beginning of Section 4 that describes the data:

All datasets analyzed in this study were collected from publicly available sources and used in accordance with their respective terms and conditions. The collection and analysis of the data comply with ethical research standards and are restricted to publicly accessible content intended for research purposes.

COMMENT 3: Your abstract cannot contain citations. Please only include citations in the body text of the manuscript, and ensure that they remain in ascending numerical order on first mention.

RESPONSE 3: We have removed citations from the abstract and checked the manuscript to ensure that citations remain in ascending numerical order.

COMMENT 4: The overall research work is good and the authors are required to address the reviewers' comments. The authors have added table for comparison of datasets and the evaluation of results. A table of comparison of literature review is also expected sharing the details, such as Ref no, Year of Research work to show the chronological order of the work, methods used, features used for analysis, name of dataset and results. Thank you

RESPONSE 4: Following the suggestion, we added the table summarizing the works on offensive language detection in Arabic, covered in Section 2.1, to the Appendix and referenced it at the end of the section. The table was placed in the Appendix to prevent its size from interrupting the flow of the related work discussion.

ADDITIONAL MODIFICATIONS:

We have ensured that dataset sizes reported in the paper are consistent.

Reviewer #1

RECOMMENDATION #1: Expand on Implicit Language: Future research should aim to explore implicit offensive language more thoroughly. As this is a significant area in offensive communication, addressing it would provide a more complete understanding of the phenomenon.

RESPONSE #1: This is indeed one of our future research directions. The generalized offensive language taxonomy will be extended to cover implicit language. This task presents greater challenges than explicit language detection, as implicit offensive language is more culturally dependent and significantly harder to identify. Additionally, the lack of dedicated Arabic datasets for implicit offensive language presents a further obstacle. To address this issue, we have added the following paragraph to the Conclusions and Limitations section:

The future work will also focus on detecting implicit offensive language in Arabic, which poses a bigger challenge due to the subtlety and context-dependency of such expressions. Moreover, the lack of implicit offensive language datasets for Arabic poses an additional challenge.

RECOMMENDATION #2: Dataset Unification: Consider developing a unified, annotated dataset that covers the wide spectrum of Arabic dialects, as this would improve the generalizability of model performance.

RESPONSE #2: Indeed, in this paper, we focused on MSA and the Levantine dialect. Other dialects that appear in publicly available datasets include Egyptian (including Gulf), Maghrebi, Darija, and Saudi, but these do not cover the full variety of Arabic dialects. The main issue with these datasets is that some contain a mixture of dialects without any dialect annotation. However, this data can be used to improve the robustness of models for offensive language detection. We have added a statement to the Conclusions section outlining this as one of our future research goals:

As part of our future work, we aim to explore the use of mixed-dialect datasets, despite their lack of dialect annotations, to enhance the robustness of models for offensive language detection across a broader range of Arabic dialects.

RECOMMENDATION #3: Fine-Tuning Insights: Offering more detailed recommendations on fine-tuning models for specific Arabic dialects would improve practical applications for offensive language detection systems.

RESPONSE #3: Following your suggestion we added these recommendations to the end of Section 5.2.2 that describes fine-tuned models performance:

Based on the results of fine-tuning for Arabic offensive language detection presented in Table 20, we suggest the following recommendations for enhancing model performance when targeting specific Arabic dialects.:

• To incorporate preprocessing techniques tailored to specific dialects, such as normalizing dialectal variations and unique morphological patterns.

• To augment datasets with synthetic examples that reflect diverse dialectal features can help mitigate data scarcity issues.

• To apply dialect-specific transfer learning by fine-tuning models on datasets that are exclusively representative of a target dialect before applying them to mixed or broader datasets could lead to more effective feature learning.

Reviewer #2

RECOMMENDATION #1: Applications and Advantages: The authors should include a paragraph in the introduction highlighting the practical applications and benefits of the proposed work. This will help contextualize its significance.

RESPONSE #1: Thank you for your suggestion. We added the following paragraph to the Introduction:

There are several real-world uses for offensive language detection in the Arabic language, such as content moderation, online community management, and social media surveillance. We strive to improve the accuracy of automated offensive detection systems by creating a uniform taxonomy and joining multiple existing datasets. Furthermore, considering the wide linguistic variety of the Arabic language in different countries, our work can facilitate the creation of dialect-specific detection models, which are currently underdeveloped.

RECOMMENDATION #2: Improved Contributions Section: The contribution paragraph should be revised to explicitly specify the techniques or added features that contribute to the improved accuracy of the proposed models.

RESPONSE #2: We rewrote the contributions part of the Introduction to make more emphasis on the advantages of our approach. This is the new contributions section:

This paper introduces a novel offensive language taxonomy tailored for Arabic, which enhances the granularity and relevance of dataset annotations (Section 3). We provide a detailed description of each level of the taxonomy, highlighting its significance in improving dataset analysis and facilitating cross-dialectal studies (Section 4). Our analysis covers 17 datasets, including re-annotated and filtered versions, and introduces a new combined dataset for Modern Standard Arabic (MSA) and Levantine dialects (Section 5.3.3).

To improve model accuracy, we evaluate pre-trained and fine-tuned Arabic transformer models, applying techniques such as transfer learning across dialects and fine-tuning on re-annotated datasets to capture nuanced patterns in offensive language (Section 5.2). Our experiments demonstrate the efficacy of combining dialect-specific datasets with the proposed taxonomy in enhancing classification performance. Finally, we discuss the broader implications of our findings for advancing research in Arabic offensive language detection and outline potential directions for future work (Section 6).

RECOMMENDATION #3: Enhanced Related Work Section: The related work section can be strengthened by connecting studies based on publication year or thematic similarities. Rather than simply referencing studies as "the paper [23]" or "the study in [42]," the authors could cite them using author names (e.g., "Levi et al.") or mention the models or approaches discussed in those studies.

RESPONSE #3: We have revised our related work section by organizing the studies based on publication year, as suggested. Also, we now cite authors explicitly, as per your recommendation.

RECOMMENDATION #4: Pictorial Representation: Including a pictorial representation of the study's steps or framework would significantly improve readers' understanding of the proposed methodology.

RESPONSE #4: Following your suggestion, we added a new Section 5.1 titled "Pipeline" to the paper, which includes a description and a figure illustrating our methodology:

The primary stages of our approach are illustrated in Figure 1. We begin by using re-annotated and filtered datasets, as detailed in Section 4.1. Next, we analyze each dataset individually, along with a combined MSA and Levantine dataset, which we construct as described in Section 5.3.3. We assess the performance of each model (outlined in Section 5.2) with and without fine-tuning on these datasets and present the results.

RECOMMENDATION #5: Results Presentation: The results section can be improved by incorporating a mix of graphical and tabular representations. Graphs can provide a visual comparison, while tables, as seen in Tables 9-19, can offer detailed numerical insights, facilitating a comprehensive performance analysis against baseline models.

RESPONSE #5: To enhance the presentation of results, we added two charts to Sections 5.3.1 and 5.3.2. These charts compare the performance of all tested models (with and without fine tuning) across all datasets, illustrating how model accuracy varies between datasets and how the models perform relative to each other. We show the charts below:

RECOMMENDATION #6: Future Work: Authors can improve the future work with more concrete research directions.

RESPONSE #6: We added two concrete research directions to our Conclusions section:

1. As part of our future work, we aim to explore the use of mixed-dialect datasets, despite their lack of dialect annotations, to enhance the robustness of models for offensive language detection across a broader range of Arabic dialects.

2. Future research can explore the adaptation of the proposed taxonomy or its core principles for offensive language detection in other languages, fostering a more inclusive online environment globally. It will also focus on detecting implicit offensive language in Arabic, which poses a bigger challenge due to the subtlety and context-dependency of such expressions. Moreover, the lack of implicit offensive language datasets for Arabic poses an additional challenge.

---

## [Decision Letter · Decision Letter 1]

23 Jan 2025

PONE-D-24-33044R1Unveiling the Spectrum of Arabic Offensive Language: Taxonomy and InsightsPLOS ONE

Dear Dr. Liebeskind,

Thank you for submitting your manuscript to PLOS ONE. After careful consideration, we feel that it has merit but does not fully meet PLOS ONE’s publication criteria as it currently stands. Therefore, we invite you to submit a revised version of the manuscript that addresses the points raised during the review process.

**Comments from PLOS Editorial Office:** We are pleased to report that the Academic Editor has evaluated your revisions and finds that no further scientific changes are necessary to proceed with publication. During internal evaluation of your manuscript by the editorial office, we noted that Table 1 of this manuscript reports examples of text that may be considered offensive. Our assessment is that these examples are not necessary for the replication or comprehension of the study, and we therefore ask you to remove this table before we proceed with publication. We appreciate your attention to this request; no other revisions are required.

We look forward to receiving your revised manuscript.

Kind regards,

Hugh Cowley

Senior Editor

PLOS One

on behalf of

Hikmat Ullah Khan, PhD (Computer Science)

Academic Editor

PLOS ONE

Journal Requirements:

Reviewers' comments:

Reviewer's Responses to Questions

**Comments to the Author**

1. If the authors have adequately addressed your comments raised in a previous round of review and you feel that this manuscript is now acceptable for publication, you may indicate that here to bypass the “Comments to the Author” section, enter your conflict of interest statement in the “Confidential to Editor” section, and submit your "Accept" recommendation.

Reviewer #2: All comments have been addressed

2. Is the manuscript technically sound, and do the data support the conclusions?

Reviewer #2: Yes

3. Has the statistical analysis been performed appropriately and rigorously? 

Reviewer #2: Yes

4. Have the authors made all data underlying the findings in their manuscript fully available?

Reviewer #2: Yes

5. Is the manuscript presented in an intelligible fashion and written in standard English?

Reviewer #2: Yes

6. Review Comments to the Author

Reviewer #2: The author has adequately addressed my comments, and the revised manuscript is now updated and suitable for publication.

7. PLOS authors have the option to publish the peer review history of their article (what does this mean?). If published, this will include your full peer review and any attached files.

Reviewer #2: No

---

## [Author Response · Author response to Decision Letter 2]

2 Feb 2025

Edits requested on your submission PONE-D-24-33044R2:

1. Please include a separate legend for each figure in your manuscript.

RESPONSE 1: We have added a separate legend for each figure.

---

## [Editor Report · Decision Letter 2]

11 Feb 2025

Unveiling the Spectrum of Arabic Offensive Language: Taxonomy and Insights

PONE-D-24-33044R2

Dear Dr. Liebeskind,

We’re pleased to inform you that your manuscript has been judged scientifically suitable for publication and will be formally accepted for publication once it meets all outstanding technical requirements.

Kind regards,

Hikmat Ullah Khan, PhD (Computer Science)

Academic Editor

PLOS ONE
---

## [Editor Report · Acceptance letter]

PONE-D-24-33044R2

PLOS ONE

Dear Dr. Liebeskind,

I'm pleased to inform you that your manuscript has been deemed suitable for publication in PLOS ONE. Congratulations! Your manuscript is now being handed over to our production team.

Kind regards,

on behalf of

Dr. Hikmat Ullah Khan

Academic Editor

PLOS ONE